# PRUNING ATTENTION HEADS WITH ALMOST-SURE SPARSITY TARGETS

## ABSTRACT

Transformer-based architectures have been widely used to obtain high accuracy values in multiple fields including natural language processing (NLP), computer vision, and more. Multi-head attention is the key factor in the success of Transformer-based architectures that has been found to be computationally expensive. Significant research effort has been devoted to improve attention compute efficiency by reducing the self-attention complexity or pruning redundant attention heads. Previous pruning work either presents training-testing inconsistency or enforces hard structural constraints which limit model performance. We propose the notion of *almost-sure* sparsity to overcome these limitations and develop a generic framework for **P**runing with **A**lmost-**S**ure **S**parsity (PASS) targets over attention heads. To further boost efficiency, we design a novel technique, *concentrator*, based on which we develop PASSCONC (**PASS** with **CONC**entrator). We investigate PASS and PASSCONC on two widely studied architectures: encoder-decoder (ED) Transformer and BERT. Experiments on IWSLT14 German-to-English translation and GLUE benchmark tasks demonstrate that our approaches outperform the SOTA by up to $1.33$ higher BLEU scores, $1.44\%$ higher accuracy, and $60\%$ higher attention layer speedups.

## 1 INTRODUCTION

Transformer-based architectures (Vaswani et al., 2017) have become a lead force in the domain of natural language processing (NLP) research, which have achieved the state-of-the-art results in a wide range of NLP tasks, including machine translation (Vaswani et al., 2017), text summarization (Liu & Lapata, 2019), and question answering (Devlin et al., 2018). More recently, significant research effort has been made to apply Transformer-based architectures to computer vision tasks including image classification (Chen et al., 2020; Dosovitskiy et al., 2020), object detection (Carion et al., 2020; Zhu et al., 2020), and video processing (Zeng et al., 2020; Zhou et al., 2018). Due to their strong representation capability (Han et al., 2020), Transformer-based architectures are found to achieve comparable or better performance than other deep learning models like CNN (LeCun et al., 1998) and RNN (Rumelhart et al., 1985).

Multi-head attention mechanism is the key factor for the high performance of Transformer-based architectures including the most powerful large language models (LLM) (Brown et al., 2020). It has been shown that multi-head attention not only helps with performance improvements (Vaswani et al., 2017), but also subject-verb agreement (Tang et al., 2018) and model interpretability analysis (Voita et al., 2019; Clark et al., 2019). However, attention computation is typically expensive and has been found to account for over $50\%$ inference latency (Wang et al., 2020a) due to its quadratic complexity and a lack of hardware optimization for its complex memory operations (e.g., splitting attention heads, reshaping and transposing key and value matrices).

Significant research effort has been devoted to improve attention inference efficiency from two orthogonal perspectives: *reducing self-attention complexity* and *pruning redundant attention heads*. As a representative work of the first stream, sparse attention (Roy et al., 2021; Tay et al., 2020; Child et al., 2019) focuses on sparsifying the attention distribution over tokens for each head to improve efficiency and head diversity. Linformer (Wang et al., 2020b) reduces the self-attention complexity from $O(N^2)$ to $O(N)$ with low-rank matrix approximation. While reducing the computation complexity of each single head, these techniques assume that all attention heads are of equal

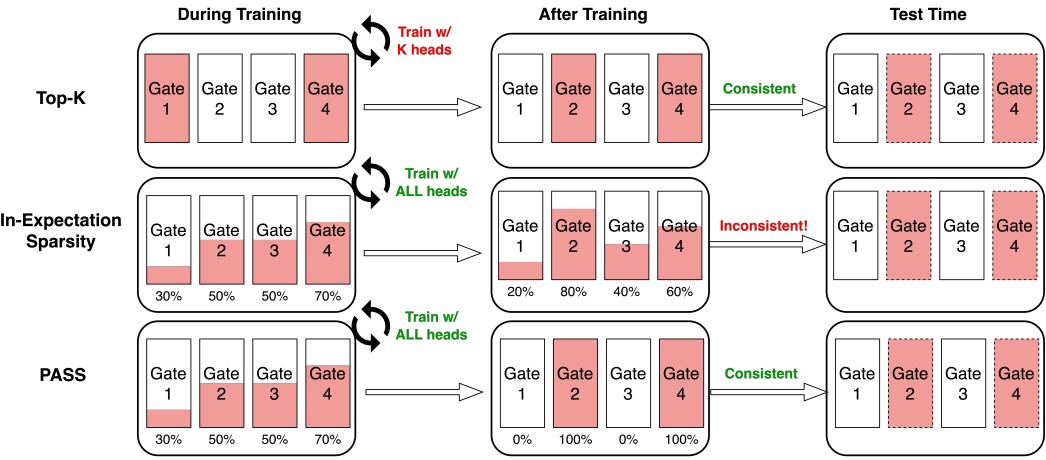

Figure 2: PASS can result in better subnetwork performance by engaging all heads in training time and consistent train-test performance by achieving *almost-sure* sparsity targets.

importance and lose the opportunity of achieving further efficiency improvements by correctly *pruning* redundant heads altogether. Recent work demonstrates the over-parameterization phenomenon in multi-head attention models. On OpenSubtitles dataset (Lison & Tiedemann, 2016), Voita et al. shows that pruning all but 4 heads out of 48 leads to a mere 0.25 drop in BLEU score. Michel et al. observes that a heuristic-based iterative approach can prune up to 40% heads from BERT, without incurring any significant performance deterioration. Moreover, by pruning 50% of all attention heads from BERT, Michel et al. observes a 17.5% speedup in inference for high batch sizes. These findings open the door for pruning attention heads from Transformer-based architectures in order to achieve efficiency improvements while maintaining high model inference performance, compared to the original unpruned model.

One line of work explores the possibility of pruning redundant attention heads from fully trained models (Michel et al., 2019). Methods like this are classified as *pipelined pruning* methods (Li et al., 2021). This paradigm identifies important attention heads from pretrained models and prunes uninformative heads subject to heuristic-based importance thresholds or rankings. Due to the separation between training and pruning, it is usually challenging for *pipelined pruning* methods to recover model capacity from pruning loss, especially in aggressive pruning settings (Li et al., 2021). In contrast, another pruning paradigm, called *joint pruning*, blends the pruning objective into the training objective and is shown to outperform

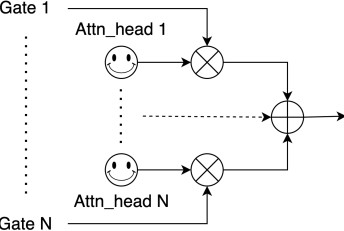

Figure 1: Illustration of *joint pruning* on multi-head attention. Gates take values from $[0, 1]$ and apply to attention heads before summation.

*pipelined pruning* by identifying subnetworks of higher performance at the same sparsity levels (Li et al., 2021). Joint pruning, as the name implies, jointly learns a set of trainable gate variables which indicate the presence of each attention head, as shown in Figure 1. Voita et al. learns the distribution for each gate and sparsifies the model by regularizing the likelihood for each gate to be open. At inference time, Voita et al. compares gate closing and opening probabilities to identify if an attention head can be pruned confidently.

A recently emerging research interest in attention head pruning is to sparsify attention heads based on user-specified sparsity targets. Li et al. iteratively applies the Gumbel-softmax trick (Gumbel, 1954) to select the top-$K$ most important attention heads for given sparsity targets. At each training iteration, only selected attention heads get updated, which limits the final model capacity from better generalization with more heads in early training stages. Xia et al. learns gate variable distributions and achieves the *target sparsity in expectation*. Though allowing all heads to participate in model training in a probabilistic manner, Xia et al. may end up with partially closed gates which leads to undesirable performance drops at test time after discretizing gate values.

Motivated by these limitations, we propose the notion of *almost-sure* sparsity, which allows us to engage all heads during model training as well as consistently end up with subnetworks of desired spar-

sity levels, as illustrated in Figure 2. We say a gate can be *closed almost surely* if the corresponding gate closing probability equals 1. In this paper, we develop PASS, a generic framework for **P**runing with **A**lmost-**S**ure **S**parsity targets by jointly learning gate distributions with the training process. To push the envelope on inference efficiency, we propose a novel technique, *concentrator*, upon which we develop PASSCONC (**PASS** with **CONC**entrator) . We evaluate our methods with encoder-decoder (ED) Transformer models and BERT models on IWSLT14 German-to-English translation (Cettolo et al., 2014) and 4 GLUE benchmark tasks (Wang et al., 2018). We explore the Pareto front between model performance and inference efficiency for subnetworks identified by PASS, PASS-CONC, and recent work (Li et al., 2021; Xia et al., 2022; Voita et al., 2019). Experiments show that PASS and PASSCONC outperform the baselines across a majority of experiment settings, by identifying subnetworks of higher speedups and comparable or better model performance. For example, on the 4 GLUE benchmark tasks, PASSCONC achieves a $185.2\%$ attention layer speedup on average that is $60\%$ higher than all baselines with even higher accuracy. This observation suggests that PASS and PASSCONC are capable of identifying subnetworks of high model capability and can be applied to resource-limited applications to achieve good performance-efficiency trade-off.

We make the following contributions.

1. We propose a novel notion of *almost-sure* sparsity to bridge the gap between real-world sparsity requirements and the probabilistic nature of recent pruning work.

2. We develop an effective model pruning framework PASS to prune models to specified *almost-sure* sparsity levels.

3. We propose a novel technique, *concentrator*, to further push the envelope on model inference efficiency and develop PASSCONC.

4. We evaluate PASS and PASSCONC on ED Transformer and BERT models with well established NLP tasks. Experiments show that PASS and PASSCONC outperform baselines by obtaining significant efficiency improvements and better performance-efficiency trade-off.

## 2 METHODOLOGY

### 2.1 GENERIC MODEL PRUNING THROUGH PROBABILISTIC GATING

A frequently encountered task in machine learning is to find the model that minimizes the negative log-likelihood of an observed dataset, which can be formulated as follows:

$$\theta^* = \arg\min_{\theta} -\log P(D|\theta) \tag{1}$$

where $D$ is an observed dataset and $\theta = \{\theta_1, \theta_2, \cdots, \theta_{|\theta|}\}$ stands for the parameters of a parameterized model (e.g., a neural network). In real-world applications, we typically have model sparsity constraints to prevent high inference latency or reduce memory footprints (Gupta & Agrawal, 2022). A recent line of work (Louizos et al., 2017; Voita et al., 2019) pursues this goal by training gate variables, $\mathbf{z} = \{z_1, z_2, \cdots, z_{|\theta|}\}$, jointly with parameters, $\theta$. Each $z_i \in \mathbf{z}$ has support $[0, 1]$. The objective function Eq. 1 can be re-parameterized as,

$$\theta^* = \arg\min_{\theta} -\log P(D|\theta \odot \mathbf{z}) \tag{2}$$

where $\odot$ indicates component-wise multiplication between network parameters $\theta$ and the gate variables $\mathbf{z}$. Typically, $\mathbf{z}$ is a latent variable following the posterior distribution $p(\mathbf{z}|D)$, which reflects the user-defined sparsity constraints. The objective function Eq. 2 becomes

$$\theta^* = \arg\min_{\theta} -\log \mathbb{E}_{p(\mathbf{z}|D)}[P(D|\theta \odot \mathbf{z})] \tag{3}$$

We aim to optimize the expected likelihood over the posterior distribution of the gate variables $\mathbf{z}$.

The objective function described by Eq. 3 is mathematically intractable when the posterior $p(\mathbf{z}|D)$ is *a priori* unknown. As an attempt to tackle such intractability, we can first derive the *evidence lower bound* of the log-likelihood in Eq. 3 which is a widely used technique in previous variational inference work (Vahdat et al., 2018b;a). Since we are interested in minimizing the negative log-likelihood, it gives us an upper bound for the objective in Eq. 3 [1],

$$-\log \mathbb{E}_{p(\mathbf{z}|D)}[P(D|\theta \odot \mathbf{z})] \leq -\mathbb{E}_{q(\mathbf{z};\Phi)}[\log P(D|\theta \odot \mathbf{z})] + KL\left(q(\mathbf{z};\Phi)||p(\mathbf{z}|D)\right) \tag{4}$$

where $q(\mathbf{z}; \Phi)$ is an *approximate posterior* distribution parameterized by $\Phi = \{\phi_1, \phi_2, \cdots, \phi_{|\theta|}\}$. Detailed derivation can be found in Appendix A.3. Minimizing this upper bound with respect to

---

[1] The posterior distribution $p$ also depends on the models but we ignore it here since it does not change the inequality.

$q(\mathbf{z}; \Phi)$ results in $q(\mathbf{z}; \Phi) = p(\mathbf{z}|D)$ and turns the inequality into an equality (Beal, 2003). By denoting this upper bound as $\mathcal{L}(\theta, \Phi)$, we can then formulate the learning problem as,

$$\mathcal{L}(\theta, \Phi) = -\mathbb{E}_{q(\mathbf{z};\Phi)}[\log P(D|\theta \odot \mathbf{z})] + KL(q(\mathbf{z};\Phi)||p(\mathbf{z}|D))$$
$$\theta^*, \Phi^* = \arg\min_{\theta, \Phi} \mathcal{L}(\theta, \Phi) \tag{5}$$

We aim to jointly learn the optimal network parameters $\theta^*$ and the distribution of gate variables, $\Phi^*$, by minimizing the upper bound $\mathcal{L}(\theta, \Phi)$.

The foregoing analysis gives a generic framework to enforce sparsity over neural models which is agnostic to the underlying network structures. To prune attention heads, all we need is to assign each head a gate variable and solve Eq. 5 with $\mathbf{z} = \{z_1, z_2, \cdots, z_{|\mathcal{H}|}\}$, where $\mathcal{H}$ is set of all attention heads. Given the generalizability of our framework, we do not explicitly distinguish attention head pruning in following analysis until Section 2.4.

## 2.2 ALMOST-SURE SPARSITY

The KL-divergence term in Eq. 5 is still mathematically intractable when the true posterior $p(\mathbf{z}|D)$ is unknown. A line of work (Voita et al., 2019) tackles this intractability by replacing the KL-divergence term with distribution-independent surrogates. A widely used surrogate (Voita et al., 2019) is $\lambda \sum_{z_i \in \mathbf{z}} Pr[z_i \neq 0]$, which can be seen as a special case of the KL-divergence term that assumes a constant ratio $\log \frac{q_\Phi(z_i)}{p(z_i|D)} = \lambda$. Though this surrogate circumvents the intractability issue, it is often challenging to identify the right $\lambda$ for a given sparsity target $s$ (Xia et al., 2022). Other work utilizes surrogates in the form of Lagrangian Multipliers (Wang et al., 2020c; Xia et al., 2022) to enforce *sparsity in expectation* for a given target. Though this approach is able to achieve target sparsities in a probabilistic manner, it may end up with partially closed gates and cause undesirable performance degradation at test time after discretizing gate values, as illustrated in Figure 3.

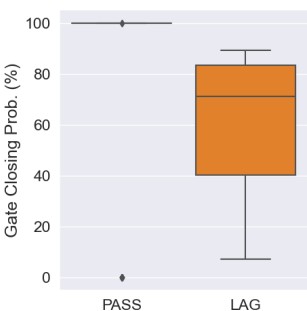

Figure 3: Lagrangian Multipliers (LAG) ends up with partially closed gates while PASS ensures almost-sure sparsity ($s = 78\%$).

In light of the limitations of previous work, we introduce the notion of *almost-sure* sparsity and propose a novel surrogate which allows us to learn empirically good approximate posteriors as well as discover subnetworks with desired target sparsities *almost surely*. The intuition behind the *almost-sure* sparsity is simple. Note that a model has sparsity $s$ provided a fraction $s$ of the gates are closed in the network. From a probabilistic perspective, it is natural to ask a subnetwork to be "confident" about which gates should be closed. In other words, gates should be closed with high probability. Mathematically, an event is said to happen almost surely, if it happens with probability 1 (Jacod & Protter, 2003). Formally, we define *almost-sure* sparsity as follows.

**Definition 1 (Almost-sure Sparsity)** *Given $s \in [0, 1)$, gate variables $\mathbf{z}$ have almost-sure sparsity $s$ if $\exists \mathbf{z}_c \subseteq \mathbf{z}$ with $|\mathbf{z}_c| = \lceil |\mathbf{z}| \cdot s \rceil$, such that $Pr[z_i$ is closed$] = 1$, $\forall z_i \in \mathbf{z}_c$ and $Pr[z_i$ is open$] = 1$, $\forall z_i \in \mathbf{z} \setminus \mathbf{z}_c$.*

We argue that the *almost-sure* sparsity is better aligned with the sparsity notion we need in static subnetworks and enables the subnetwork discovery with desired sparsity targets. Next, we present learning objectives designed to achieve *almost-sure* sparsity targets specified by users.

## 2.3 LEARNING OBJECTIVE WITH ALMOST-SURE SPARSITY

We aim to learn a good approximate posterior $q(\mathbf{z}; \Phi)$ with desired almost-sure sparsity. In this paper, we adopt the Hard Concrete distribution (Louizos et al., 2018) as the basic form of the approximate posterior $q(\mathbf{z}; \Phi)$, given its continuous-discrete nature and its wide application in model pruning (Voita et al., 2019; Xia et al., 2022).

Hard Concrete distribution has its support over the closed interval $[0, 1]$ and non-zero probability mass at 0 and 1. Hard Concrete distribution is derived by stretching and collapsing the Concrete distribution (Maddison et al., 2016), as illustrated in Figure 4 (left). We introduce derivation details in Appendix A.1. For each gate $z_i \in [0, 1]$ following Hard Concrete distribution, the corresponding probability mass at 0 and 1 with respect to $q(z_i; \phi_i)$ are given as $q(z_i = 0; \phi_i) = \text{sig}\left(\beta \log\left(\frac{-\gamma}{\zeta}\right) - \phi_i\right)$, $q(z_i = 1; \phi_i) = \text{sig}\left(\phi_i - \beta \log\left(\frac{1-\gamma}{\zeta-1}\right)\right)$. For simplicity of notation, we

denote $q_0(\phi_i) := q(z_i = 0; \phi_i)$, the gate *closing* probability, and $q_1(\phi_i) := q(z_i = 1; \phi_i)$, the gate *opening* probability. Due to the monotonicity of the sigmoid function, when $\phi_i$ increases, $q_1(\phi_i)$ increases and $q_0(\phi_i)$ decreases, and gate $z_i$ is more likely to open. We further define $q_{nb}(\phi_i) = 1 - q_0(\phi_i) - q_1(\phi_i)$ as the probability for $z_i$ being *non-binary*. We use $\beta = 0.33$, $\gamma = -0.1$, and $\zeta = 1.1$ by default, following previous work (Voita et al., 2019). Clearly, the closing and opening probability of each $z_i \in \mathbf{z}$ are differentiable functions of $\phi_i \in \Phi$, as shown in Figure 4 (right). By jointly learning $\Phi$ with the network parameters, we are able to almost-surely close (resp. open) gates $z_i \in \mathbf{z}$ by continuously increasing (resp. decreasing) the values of $\phi_i \in \Phi$, using gradient-descent optimizers. At each training iteration, gates are sampled w.r.t. the learnt distribution and then applied to attention heads to achieve pruning.

At the end of pruning, we want $q(\mathbf{z}; \Phi)$ to achieve almost-sure sparsity for a given target $s$. Our strategy is to design a learning objective that meets the desired almost-sure sparsity at its optimum, and optimize it along with model training. It is worth pointing out that there exists a family of learning objectives satisfying this criterion. However, not all of them can be easily optimized to their minimum, especially by gradient descent optimizers (Kingma & Ba, 2015). For example, one may propose to minimize the following objective.

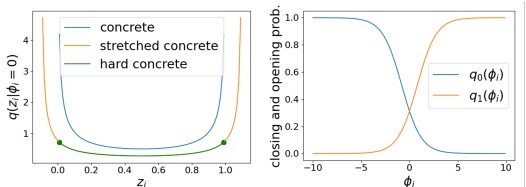

Figure 4: (left) Hard concrete distribution derived by streching-and-collapsing a Concrete distribution. (right) Closing and opening probability of gating variables are differentiable functions of $\phi_i$. $\beta = 0.33$, $\gamma = -0.1$, $\zeta = 1.1$.

$$\mathcal{R}_{base}(\Phi, s) = \sum_{i=1}^{|\theta|} q_{nb}(\phi_i) + \left| s|\theta| - \sum_{i=1}^{|\theta|} q_0(\phi_i) \right| \tag{6}$$

It can be easily seen that $\mathcal{R}_{base}$ takes on its minimum value 0 when achieving almost-sure sparsity $s$. However, there exist local optima that may prevent gradient descent optimizers from converging to the global optimum. To illustrate this, for simplicity, we visualize the values of $\mathcal{R}_{base}$ in a 2-gates setting $\mathbf{z} = \{z_1, z_2\}$ in Figure 5. With 2 gates and a sparsity target $s = 0.5$, we want one gate to be almost-surely closed and the other gate almost-surely open. In Figure 5, such global optima correspond to the top-left and bottom-right corner where one of $\phi_1$ or $\phi_2$ takes on a high value and the other takes on a low value. However, it can be clearly observed that there exist a local optimum in the top-right region which corresponds to the situation where both gates are open with high probability. In other words, with $\mathcal{R}_{base}$, if both $\phi_1$ and $\phi_2$ happen to take positive values due to noise from the training process or bad initialization, the gradient descent direction will increase the probability for both gates to be open and fail to meet the sparsity target $s = 0.5$. Under weak conditions[2], we can prove that the gradient descent direction of $\mathcal{R}_{base}$ always leads to a higher opening probability for gate $z_i$ if $\phi_i \geq \log(\frac{-1-\sqrt{1-g(a)g(-a)}}{g(a)})$, where $g(a) = 2e^a - e^{-a}$, $a = \beta \log(\frac{-\gamma}{\zeta})$.

In light of the limitation of $\mathcal{R}_{base}$, we propose the following learning objective,

$$\mathcal{R}_{pass}(\Phi, s) = \sum_{i=1}^{|\theta|} q_{nb}(\phi_i) + \left| s|\theta| - \sum_{i=1}^{|\theta|} q_0(\phi_i) \right| + \left| (1-s)|\theta| - \sum_{i=1}^{|\theta|} q_1(\phi_i) \right| \tag{7}$$

$\mathcal{R}_{pass}$ does not suffer from the local optimality issue that $\mathcal{R}_{base}$ does, as illustrated in Figure 5. In fact, we can show that minimizing $\mathcal{R}_{pass}$ always generates neither over-sparse nor over-dense subnetworks. In order to show this formally, define the *expected density* as $\frac{1}{|\theta|} \sum_{i=1}^{|\theta|} q_1(\phi_i)$, and the *expected sparsity* as $\frac{1}{|\theta|} \sum_{i=1}^{|\theta|} q_0(\phi_i)$. We have the following lemma.

**Lemma 1** *Minimizing $\mathcal{R}_{pass}$ always leads to a subregion in the search space where the expected sparsity is no more than $s$ and expected density is no more than $1 - s$.*

---

[2]We assume the Hard Concrete distribution is equally stretched in both directions, which gives $\gamma + \zeta = 1$.

Proof can be found in Appendix A.2. By substituting the KL-divergence term in Eq. 5 with $\mathcal{R}_{pass}$, we obtain the PASS optimization objective where $\lambda$ is the regularization coefficient.

$$\mathcal{L}_{pass}(\theta, \Phi) = -\mathbb{E}_{q(\mathbf{z};\Phi)}[\log P(D|\theta \odot \mathbf{z})] + \lambda \mathcal{R}_{pass}(\Phi, s)$$
$$\theta_{pass}, \Phi_{pass} = \arg\min_{\theta, \Phi} \mathcal{L}_{pass}(\theta, \Phi) \tag{8}$$

### 2.4 CONCENTRATOR

To further improve model inference efficiency, we propose the use of concentrator. Wang et al. observed that the auxiliary operations in multi-head attention computation (e.g., reshaping and transposing matrices, heads splitting, and concatenation) account for $73\%$ of the overall latency in attention layers. The run-time overhead can hardly be avoided as long as there exist unpruned heads in the attention layers. Con-

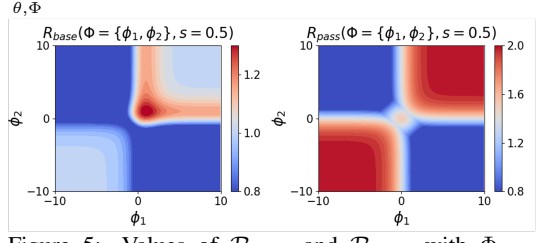

Figure 5: Values of $\mathcal{R}_{base}$ and $\mathcal{R}_{pass}$ with $\Phi = \{\phi_1, \phi_2\}$ and $s = 0.5$.

sider subnetworks of the same attention head sparsity, intuitively, if the unpruned attention heads are inclined to *concentrate* among a few layers, the other layers can be entirely skipped, saving the run-time overhead and improving inference efficiency. Given this, we propose the *concentrator* to encourage the unpruned attention heads to be concentrated on as few layers as possible.

Given a Transformer-based model of $L$ layers and $H$ heads per layer, the concentrator is defined as $\mathcal{R}_{conc}(\Phi) = \sum_{l=1}^{L} \left(1 - \prod_{h=1}^{H} q_0(\phi_{l,h})\right)$, where $\phi_{l,h}$ is the distribution parameter for the $h$-th gate variable on the $l$-th layer. Notice that $1 - \prod_{h=1}^{H} q_0(\phi_{l,h})$ indicates if the $l$-th layer can be entirely skipped: it takes on a value 0 only if all heads of the layer have a closing probability 1. $\mathcal{R}_{conc}$ is a summation of the layer-wise indicators over all layers and has regularization effects by penalizing the levels of unconcentration. We introduce $\lambda_c$ to control the concentrator effects and obtain the following optimization objective for PASSCONC, i.e., PASS with concentrator.

$$\mathcal{L}_{passconc}(\theta, \Phi) = -\mathbb{E}_{q(\mathbf{z};\Phi)}[\log P(D|\theta \odot \mathbf{z})] + \lambda \mathcal{R}_{pass}(\Phi, s) + \lambda_c \mathcal{R}_{conc}(\Phi)$$
$$\theta_{passconc}, \Phi_{passconc} = \arg\min_{\theta, \Phi} \mathcal{L}_{passconc}(\theta, \Phi) \tag{9}$$

In practice, with proper training settings, the proposed approach can discover subnetworks with the desired sparsities and high accuracy. Note that we approach almost sure sparsity by increasing or decreasing $\phi_i \in \Phi$ with gradient-descent optimizers. However, as $\phi_i$'s get polarized, their gradients gradually converge to 0 as illustrated in Figure 4 (right). Consequently, gates closed (resp. opened) with high probability in early training stage are unlikely to be self-adaptively re-opened (resp. closed) in later training iterations, which may lead to sub-optimal pruning results. We propose to resolve this issue with a clipping and selective reopening strategy. The idea of clipping is widely used in training deep learning models to avoid gradient exploding and vanishing (Zhang et al., 2019; Koloskova et al., 2023).In this same spirit, we clip $\phi_i$ to predefined ranges to alleviate the aforementioned issues caused by small gradients.[3] To further incentivize training dynamics, we propose to selectively reopen closed gates with respect to the gate quality. There is a line of work on how to measure gate qualities, including importance score (Michel et al., 2019), head confidence (Voita et al., 2019), LPR (Ding et al., 2017), etc. We choose head confidence because it is found to be an informative heuristic notion (Behnke & Heafield, 2020) and requires little to no additional computation. The confidence of a head is the average maximum attention weights of tokens in a set of sentences (Voita et al., 2019). We normalize confidence scores for each attention head and reopen almost-surely closed gates[4] with a probability equal to the normalized scores.

## 3 EXPERIMENTS

### 3.1 EXPERIMENTAL SETUP

#### 3.1.1 MODEL, DATA, AND METRICS

**Model**: We investigate two widely-used Transformer-based architectures: encoder-decoder (ED) Transformer (Vaswani et al., 2017) and BERT (Devlin et al., 2018). We use the FAIRSEQ toolkit (Ott et al., 2019) to implement a 6-layer ED Transformer with 72 heads in total, and the HUGGING FACE codebase (Wolf et al., 2020) to implement a 12-layer BERT with 144 heads in total.

---

[3]In our implementation, we empirically choose $[-5, 5]$ as the clipping range.
[4]To reopen an almost-surely closed gate $z_i$, we manually decrease its closing probability by increasing $\phi_i$.

**Datasets**: Following previous work (Li et al., 2021), the ED Transformer model is trained and evaluated on the IWSLT14 German-to-English translation dataset (Cettolo et al., 2014). The BERT model is fine-tuned and evaluated on 4 benchmark NLP tasks from the GLUE benchmark (Wang et al., 2018) including the Multi-Genre Natural Language Inference (MNLI) dataset (Williams et al., 2018), the Question-answering NLI (QNLI) dataset (Rajpurkar et al., 2016), the Quora Question Pairs (QQP) dataset (Sharma et al., 2019), and the Stanford Sentiment Treebank (SST-2) dataset (Socher et al., 2013).

**Metrics**: We use BLEU score (Papineni et al., 2002) as the metric to measure model performance on the translation task following previous work (Li et al., 2021; Michel et al., 2019), and use accuracy as the metric on the 4 GLUE benchmark tasks following Wang et al. (2018). In addition, we are also interested in the efficiency improvements achieved by PASS and PASSCONC. We use wall clock time to measure the efficiency w.r.t. latency.

### 3.1.2 BASELINES

We consider three strong baselines that prune attention heads to a specified sparsity level.

**Voita et al. (2019) (Voita)** Voita et al. (2019) follows the *joint pruning* paradigm and prunes attention heads by applying the stochastic approximation to $L_0$ regularization (Louizos et al., 2018) to gate closing probabilities. Voita et al. (2019) achieves pruning by jointly training models with the following regularization term $\mathcal{R}_{voita}(\Phi) = \lambda_v \sum_{h=1}^{|\mathcal{H}|}(1 - q_0(\phi_h))$, where $\lambda_v$ can be used to *indirectly* control the achieved sparsities [5].

**Differentiable Subset Pruning (DSP)** DSP Li et al. (2021) applies the Gumbel-softmax trick (Gumbel, 1954) to select the top-$K$ attention heads for a given sparsity target. DSP learns a $K$-hot vector $g_h$ by iteratively applying Gumbel-softmax $K$ times, where $g_h = \sum_{k=1}^{K} g_h^k = \sum_{k=1}^{K} \frac{exp(r_h^k/\tau)}{\sum_{h'=1}^{H} exp(r_{h'}^k/\tau)}$, $r_h^k = r_h^{k-1} + \log(1 - g_h^{k-1})$, and $r_h^1 = w_h + n_h$. $w_h$ denotes trainable parameter indicating head importance, $n_h \sim Gumbel(0, 1)$ is Gumbel noise, and $\tau$ is a hyper-parameter that controls the annealing temperature.

**Lagrangian Multiplier (LAG)** A recent line of work (Xia et al., 2022; Wang et al., 2020c) employs Lagrangian Multiplier (Wang et al., 2020c) to enforce *sparsity in expectation*. Given a sparsity target $s$, LAG trains models along with the regularization term $\mathcal{R}_{lag} = \lambda_1(\hat{s} - s) + \lambda_2(\hat{s} - s)^2$, where $\hat{s}$ is the *expected sparsity*. $\lambda_1$ and $\lambda_2$ are trainable parameters and will be optimized jointly in training.

### 3.1.3 PROTOCOLS

We express sparsity targets over attention heads $\mathcal{H}$ interchangeably as $s \in (0, 1)$ and as integer $K$ where $K = \lfloor(1 - s)|\mathcal{H}|\rfloor$, the number of unpruned heads. Unless stated otherwise, for a given sparsity target $K$, we evaluate all methods by selecting the top-$K$ most important heads w.r.t. the corresponding ranking metrics, i.e., the gate opening probabilities for PASS, PASSCONC, Voita, and LAG, and the head importance score $w_h$ for DSP. Detailed hyper-parameter settings are in Appendix A.4. We test all methods on both architectures with target tasks (30 training epochs for ED Transformer; 3 fine-tuning epochs for BERT as in Li et al.). All experiments are conducted on a high performance compute cluster equipped with NVIDIA P100 GPUs (each with 12GB GPU RAM). All codes will be released through GitHub after reviews.

## 3.2 PASS AND PASSCONC IMPROVE MODEL PERFORMANCE

We investigate the model performance of subnetworks identified by PASS, PASSCONC, LAG, DSP, and Voita under various sparsity constraints. We compare all five methods on both ED Transformer and BERT models. The results are summarized in Table 1. More results are in Appendix A.5.

On IWSLT14 German-to-English translation task, PASS and PASSCONC outperform all 3 baselines in a majority of sparsity settings. When $K = 16$, both PASS and PASSCONC achieve BLEU scores of $\sim 32.7$, which is $\sim 1.3$ higher than DSP $\sim 1.8$ higher than LAG, and $\sim 5.2$ higher than Voita. On the 4 GLUE benchmark tasks, we observe a similar trend in high sparsity situations. When $K = 16$, PASS and PASSCONC achieve average model accuracy of $86.27\%$ and $85.25\%$ respectively, while DSP drops to $84.47\%$, LAG drops to $83.84\%$, and Voita drops to $84.83\%$. When sparsity targets are

---

[5]We use the recommended $\lambda_v$ values from (Li et al., 2021) for each sparsity setting.

| | BLEU(IWSLT14) | | | | | AVG_Accuracy(MNLI, QQP, QNLI, SST-2) | | | | |
|---|---|---|---|---|---|---|---|---|---|---|
| K | PASS | PASSCONC | DSP | LAG | Voita | PASS | PASSCONC | DSP | LAG | Voita |
| 16 | **32.73** | 32.70 | 31.40 | 30.91 | 27.55 | **86.27** | 85.25 | 84.47 | 83.84 | 84.83 |
| 32 | 33.45 | **33.48** | 33.42 | 32.66 | 32.80 | 87.59 | 86.47 | **88.36** | 86.99 | 87.15 |
| 48 | 33.89 | 33.91 | **34.00** | 33.12 | 32.97 | **88.65** | 88.30 | 88.52 | 88.02 | 88.02 |
| 64 | 34.01 | **34.05** | 33.89 | 33.02 | 33.20 | 88.72 | 88.62 | **88.81** | 88.40 | 84.20 |

Table 1: Subnetwork performance on IWSLT14 De-En translation task and GLUE benchmark tasks.

| | Speedup(IWSLT14) (%) | | | | | AVG_Speedup(MNLI, QQP, QNLI, SST-2) (%) | | | | |
|---|---|---|---|---|---|---|---|---|---|---|
| K | PASS | PASSCONC | DSP | LAG | Voita | PASS | PASSCONC | DSP | LAG | Voita |
| 16 | 144.3 | **162.8** | 141.1 | 141.1 | 142.7 | 114.4 | **185.2** | 123.1 | 120.4 | 126.1 |
| 32 | 115.5 | **118.7** | **118.7** | 110.4 | 117.6 | 107.1 | **135.6** | 107.3 | 105.4 | 105.3 |
| 48 | 101.6 | 104.1 | **105.8** | 102.4 | 102.4 | 103.1 | **109.3** | 102.7 | 102.9 | 103.8 |
| 64 | 100.8 | **104.1** | 100.0 | 100.0 | 100.0 | 103.2 | **106.4** | 102.9 | 103.0 | 103.0 |

Table 2: Attention layer speedups on IWSLT14 De-En translation task and GLUE benchmark tasks.

low, PASS is able to match or outperform all 3 baselines, while PASSCONC can be outperformed by the strongest baseline DSP while still being comparable to the remaining two.

One interesting observation is that Voita delivers surprisingly low accuracy in low sparsity settings (e.g., $K = 64$) with GLUE benchmark tasks. The degraded performance can be attributed to its intrinsic sensitivity to the choice of $\lambda_v$, which is used to indirectly control sparsity targets. Li et al. (2021) observed that a small increase in $\lambda_v$ (e.g., $0.0009 \rightarrow 0.0014$) may lead to drastic change of achieved sparsity (e.g., the number of unpruned heads decreases from 30 to 11), which suggests that Voita is inadequate when users require subnetworks of pre-defined number of attention heads.

### 3.3 PASSCONC IMPROVES MODEL EFFICIENCY

We evaluate the attention layer speedups for subnetworks identified under various sparsity constraints, at inference time. We report the inference speedups in comparison to the unpruned model. The results are summarized in Figure 6 and Table 2. More results can be found in Appendix A.5.

On the 4 GLUE benchmark tasks with BERT models, PASSCONC outperforms all baselines across a majority of sparsity constraints with great efficiency improvements and comparable or better accuracy. When $K = 16$, PASSCONC achieves a 185.2% speedup, which is $\sim 60\%$ higher than all baselines, and an average accuracy 85.25% that is also higher than DSP, LAG, and Voita. PASS has a better accuracy but a lower speedup. As the sparsity targets decrease (i.e, as $K$ increases), the speedups achieved by all methods in general goes down but PASSCONC always dominates the competition in terms of efficiency, at the price of a relatively small drop in performance. On IWSLT14 German-to-English task with ED Transformer model, PASSCONC outperforms all baseline methods in almost all sparsity settings, (see Table 2). When $K = 16$, PASSCONC achieves a 162.8% speedup, which is more than 20% higher than all baselines, with at least 1.3 higher BLEU scores.

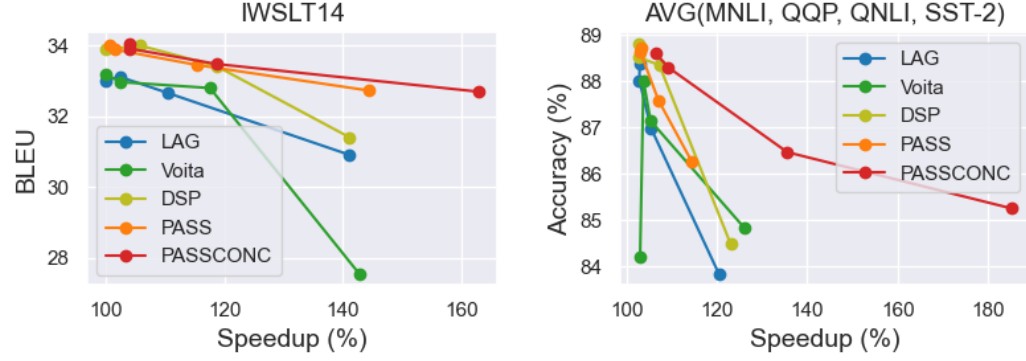

Figure 6: Attention layer speedups v.s. subnetwork performance on IWSLT14 De-En translation task and GLUE benchmark tasks.

### 3.4 ABLATION STUDY

Previous analysis of PASS and PASSCONC demonstrates the significant efficiency improvements brought about by the concentrator (see §2.4). We validate that the clipping and reopening strategy is necessary for PASSCONC to obtain significant efficiency improvements. As shown in Figure 7, without the clipping and reopening strategy, the speedups achieved by PASSCONC can reduce by up to 70%! This observation demonstrates the necessity of dynamically re-activating closed gates to help model converge to cost-effective regions, as desired by concentrator.

## 4 RELATED WORK

*Unstructured pruning* has been well studied in the literature (Gupta & Agrawal, 2022) and dates back to Optimal Brain Damage (LeCun et al., 1989). Unstructured pruning prunes individual parameters and identifies subnetworks of high sparsity. However, unstructured pruning hardly achieves practical efficiency improvements without specialized software and hardware supports (Xia et al., 2022). In contrast, *structured pruning* prunes groups of parameters within certain structures (e.g., channels and attention heads). Structured pruning has been widely explored in computer vision tasks (He et al., 2017) and has started to attract research interest in the NLP community. Research efforts have been devoted to designing pruning strategies at both coarse- and fine-grained levels (Xia et al., 2022; Prasanna et al., 2020) over structures like feed-forward layers and attention heads. Previous work on attention head pruning (Li et al., 2021; Michel et al., 2019; Voita et al., 2019) either presents a lack of training-testing consistency by focusing on *sparsity in expectation*, or leads to limited model capacity due to hard structural constraints. We focus on structured pruning and propose the notion of *almost-sure* sparsity to overcome the above limitations.

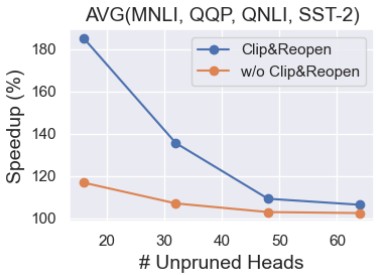

Figure 7: Ablation study on the clipping and reopening strategy.

In addition to pruning, many other techniques have been developed to obtain inference efficiency for deep learning models. Other than sparsity over the number of attention heads, a line of work (Roy et al., 2021; Child et al., 2019) focuses on sparsifying the attention distribution over tokens for each head to improve efficiency and head diversity. Recently, Correia et al. and Treviso et al. propose to adaptively sparsify the attention distribution by enforcing low attention scores to be exactly 0 through $\alpha$-entmax (Peters et al., 2019). Wang et al. develops efficient self-attention design of linear complexity by using low-rank matrix approximation. Other techniques include quantization (Shen et al., 2020), knowledge distillation (Hinton et al., 2015), parameter sharing (Ling et al., 2015), tensor decomposition (Oseledets, 2011) etc. We refer interested readers to (Gupta & Agrawal, 2022; Treviso et al., 2022) for a comprehensive survey.

## 5 DISCUSSION AND CONCLUSION

We propose a novel notion of *almost-sure* sparsity, develop a generic framework for **P**runing with **A**lmost-**S**ure **S**parsity (PASS) targets, and demonstrate its pruning capacity with attention heads. To further push the envelope on inference efficiency, we propose a novel technique, concentrator, based on which we develop PASSCONC (**PASS** with **CONC**entrator). We investigate PASS and PASSCONC on two widely studied architectures: encoder-decoder (ED) Transformer and BERT. Experiments on IWSLT14 German-to-English translation and 4 GLUE benchmark tasks (Wang et al., 2018) demonstrate that PASS and PASSCONC outperform the SOTA methods DSP, LAG, and Voita by identifying subnetworks of up to 1.33 higher BLEU scores, 1.44% higher accuracy, and 60% higher speedups, at the same sparsity levels.

We conclude that PASS and PASSCONC can be used to identify high performance subnetworks and help address the challenge of deploying large language models in resource-limited applications. In the future, we would like to explore the possibility of extending the proposed framework to multiple model structures (e.g., feed-forward layers) and prune for meeting other target footprint metrics such as latency and memory, in addition to sparsity. Also, since both PASS and PASSCONC are agnostic to the underlying self-attention implementation, it is intriguing to investigate the compound efficiency improvements achieved by combining our approaches with other efficiency techniques such as linear multi-attention (Wang et al., 2020b).

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

## A APPENDIX

### A.1 HARD CONCRETE DISTRIBUTION (LOUIZOS ET AL., 2018)

Hard Concrete distribution is derived from the Concrete distribution (Maddison et al., 2016). The cumulative distribution function of a binary Conrete random variable $z_i^c \in (0, 1)$ is as follows:

$$Q_c(z_i^c; \beta, \phi_i) = \text{sig}\left(\left(\log z_i^c - \log(1 - z_i^c)\right) \beta - \phi_i\right) \quad (10)$$

where $\text{sig}(x) = \frac{1}{1+e^{-x}}$ is the sigmoid function and $0 < \beta < 1$ is a constant. We can obtain the Hard Concrete distribution by first stretching the support of the Concrete distribution from $(0, 1)$ to $(\gamma, \zeta)$, where $\gamma < 0$, $\zeta > 1$, and then collapsing the probability mass on $(\gamma, 0]$ (resp. $[1, \zeta)$) to the endpoint 0 (resp. 1). Equivalently, for each Concrete random variable $z_i^c \in (0, 1)$, we can obtain a Hard Concrete random variable $z_i = \min(1, \max(0, z_i^c(\zeta - \gamma) + \gamma)$. For each $z_i$, the closing probability $q(z_i = 0; \phi_i) = Pr[z_i^c \leq \frac{-\gamma}{\zeta-\gamma}] = Q_c(\frac{-\gamma}{\zeta-\gamma}; \beta, \phi_i)$, and the opening probability $q(z_i = 1; \phi_i) = Pr[z_i^c \geq \frac{1-\gamma}{\zeta-\gamma}] = 1 - Q_c(\frac{1-\gamma}{\zeta-\gamma}; \beta, \phi_i)$. By plugging in Eq. 10, we have the gate closing and opening probabilities.

## A.2 PROOF OF LEMMA 1

We include the proof of Lemma 1 as follows.

**Proof**: We prove this lemma by showing that, in every possible case, the gradient descent direction of the objective $\mathcal{R}_{pass}$ always leads to a subregion in the search space where the expected sparsity is no more than $s$ and expected density is no more than $1 - s$. Recall that we define *expected density* as $\frac{1}{|\theta|} \sum_{i=1}^{|\theta|} q_1(\phi_i)$, and the *expected sparsity* as $\frac{1}{|\theta|} \sum_{i=1}^{|\theta|} q_0(\phi_i)$. For the simplicity of proof language, we use $E_d$ to denote the expected density and $E_s$ to denote the expected sparsity. Intuitively, as the expected density increases (resp. decreases), the expected sparsity will monotonically decrease (resp. increase). One important observation is that, *the sum of expected density and sparsity is always less than* 1, due to the fact that $q_1(\phi_i) + q_0(\phi_i) = 1 - q_{nb}(\phi_i)$ and $q_{nb}(\phi_i) > 0$.

Given a sparsity target $s$, depending on the values of expected density and sparsity, there are three possible situations: (i) the easiest case is that **the model is already neither over-sparse nor over-dense** (i.e., $E_d \leq 1 - s$ and $E_s \leq s$). In this case, we can rewrite $\mathcal{R}_{pass}(\Phi, s) = 2 \sum_{i=1}^{|\theta|} q_{nb}(\phi_i)$. Minimizing $\mathcal{R}_{pass}(\Phi, s)$ amounts to minimizing $q_{nb}(\phi_i)$, which polarizes the gates to achieve the required almost-sure sparsity, until either $E_d = 1 - s, E_s < s$ or $E_d < 1 - s, E_s = s$ which is aligned with Lemma 1. (ii) **high expected density and low expected sparsity** (i.e., $E_d \geq 1 - s$ and $E_s \leq s$), in which case we can rewrite Eq. 7 to obtain $\mathcal{R}_{pass}(\Phi, s) = \sum_{i=1}^{|\theta|} q_{nb}(\phi_i) + s|\theta| - \sum_{i=1}^{|\theta|} q_0(\phi_i) - (1 - s)|\theta| + \sum_{i=1}^{|\theta|} q_1(\phi_i)$. Because $q_{nb}(\phi_i) + q_1(\phi_i) = 1 - q_0(\phi_i)$, we can further simplify $\mathcal{R}_{pass}(\Phi, s)$ to be $\mathcal{R}_{pass}(\Phi, s) = 2 \sum_{i=1}^{|\theta|} \left( s - q_0(\phi_i) \right)$. Clearly, minimizing $\mathcal{R}_{pass}(\Phi, s)$ amounts to increasing $q_0(\phi_i)$ that leads to lower expected density and higher expected sparsity until $E_d = 1 - s, E_s < s$, which is aligned with Lemma 1; (iii) **low expected density and high expected sparsity** (i.e., $E_d \leq 1 - s$ and $E_s \geq s$). Similarly, we can rewrite $\mathcal{R}_{pass}(\Phi, s) = 2 \sum_{i=1}^{|\theta|} \left( 1 - s - q_1(\phi_i) \right)$. Minimizing $\mathcal{R}_{pass}(\Phi, s)$ amounts to increasing $q_1(\phi_i)$ that brings down expected sparsity as well as increases expected density until $E_d < 1 - s, E_s = s$, which is aligned with Lemma 1; It is impossible to have both **high expected density and expected sparsity** due to $E_s + E_d < 1$. We have shown that Lemma 1 holds in all possible cases. $\square$

## A.3 DERIVATION OF THE UPPER BOUND IN EQ. 4

Here we present how to derive the upper bound in Eq. 4.

$$-\log \mathbb{E}_{p(\mathbf{z}|D)}[P(D|\theta \odot \mathbf{z})]$$

$$= -\log \int_{\mathbf{z}} p(\mathbf{z}|D) P(D|\theta \odot \mathbf{z})$$

$$= -\log \int_{\mathbf{z}} p(\mathbf{z}|D) P(D|\theta \odot \mathbf{z}) \frac{q_\Phi(\mathbf{z})}{q_\Phi(\mathbf{z})}$$

$$= -\log \mathbb{E}_{q_\Phi(\mathbf{z})} \left[ P(D|\theta \odot \mathbf{z}) \frac{p(\mathbf{z}|D)}{q_\Phi(\mathbf{z})} \right] \qquad (11)$$

$$\leq -\mathbb{E}_{q_\Phi(\mathbf{z})} \left[ \log \left( P(D|\theta \odot \mathbf{z}) \frac{p(\mathbf{z}|D)}{q_\Phi(\mathbf{z})} \right) \right]$$

$$= -\mathbb{E}_{q_\Phi(\mathbf{z})}[\log P(D|\theta \odot \mathbf{z})]$$
$$+ KL\left( q_\Phi(\mathbf{z}) \| p(\mathbf{z}|D) \right)$$

## A.4 HYPER-PARAMETER SETTINGS

We adopt an exponential escalation strategy to increase the value of regularization coefficient $\lambda$.

$$\lambda = \lambda_{base} \cdot \lambda_0^{\#\text{train\_itr} / \#\text{n\_step}} \qquad (12)$$

where $\lambda_{base}$ and $\lambda_0$ are hyper-parameters. We choose #n_step as $1,000$ in all experiments. As for PASSCONC, $\lambda_c$ is defined as the minimal ratio between sparsifier gradients and concentrator gradients of $\phi_i \in \Phi$ for all heads to ensure that the concentrator is not dominated by the sparsifier

|  | ED Transformer | BERT |
|---|---|---|
| $\lambda_{base}$ | 1 | 1e-5 |
| $\lambda_0$ | 2 | 1000 |
| learning rate for $\Phi$ | 0.2 | 0.5 |

Table 3: Hyper-parameters

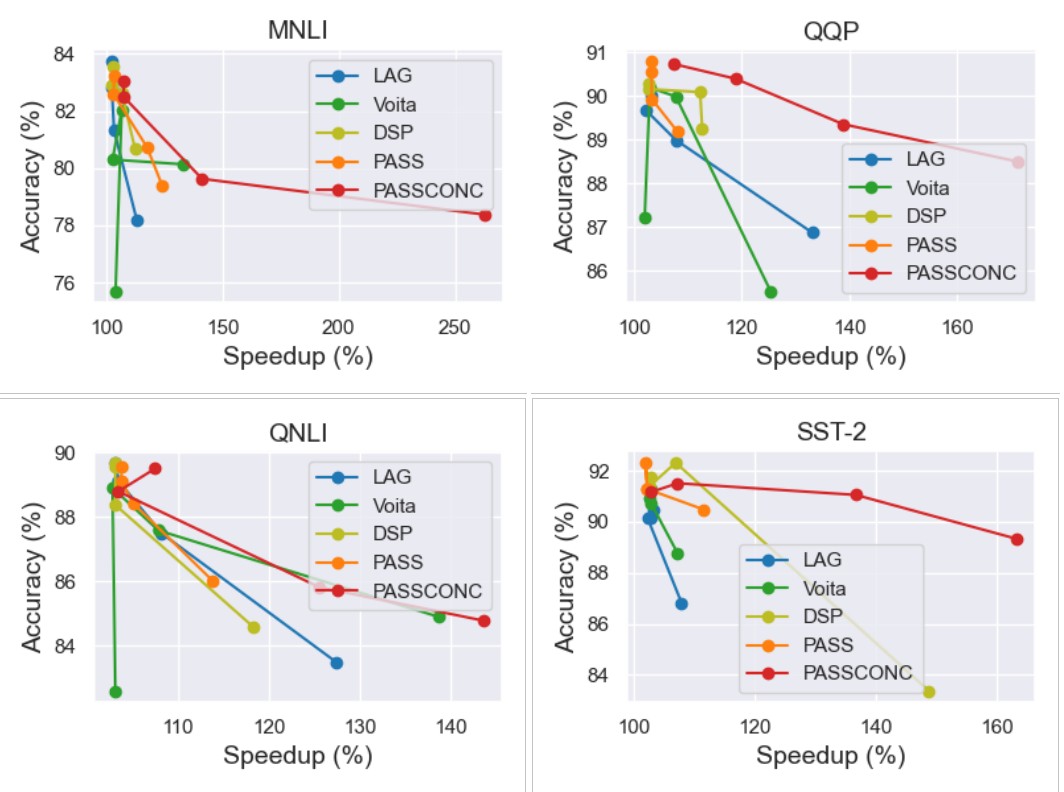

Figure 8: Attention layer speedups v.s. subnetwork performance on MNLI, QQP, QNLI, and SST-2.

while the concentration effect is modest to avoid damaging model training quality.

$$\lambda_c = \lambda \cdot \min\left\{ \left|\frac{\partial \mathcal{R}_{pass}}{\partial \phi_i}\right| \Big/ \left|\frac{\partial \mathcal{R}_{conc}}{\partial \phi_i}\right| \Big| \phi_i \in \Phi \right\} \tag{13}$$

$\lambda_c$ is adaptively updated in the middle of training, and set to 0 in both the early training phase and the last few training iterations to help improve model performance and achieve sparsity convergence. We use the same inverse square root learning rate schedules for model training as in Li et al..

We use the following hyper-parameters across all our experiments, as summarized in Table 3. For experiments involving concentrators, $\lambda_c$ is set to 0 during the first $20,000$ iterations and the last $7,000$ iterations when training ED Transformer models. For BERT models, $\lambda_c$ is set to 0 except for iterations between $2,000$ and $5,000$.

## A.5  EFFICIENCY AND PERFORMANCE EVALUATION ON MNLI, QQP, QNLI, AND SST-2

In this section, we provide detailed speedup and performance results on each of the 4 GLUE benchmark tasks, as shown in fig. 8 and tables 4 to 7.

| | Speedup(MNLI) (%) | | | | | Speedup(QQP) (%) | | | | |
|---|---|---|---|---|---|---|---|---|---|---|
| K | PASS | PASSCONC | DSP | LAG | Voita | PASS | PASSCONC | DSP | LAG | Voita |
| 16 | 124.2 | **262.7** | 112.5 | 113.2 | 132.9 | 108.2 | **171.2** | 112.7 | 133.1 | 125.4 |
| 32 | 117.7 | **141.3** | 106.7 | 103.5 | 102.8 | 103.4 | **138.8** | 112.3 | 107.8 | 107.8 |
| 48 | 103.2 | **107.4** | 102.5 | 102.5 | 106.7 | 103.4 | **118.9** | 102.7 | 102.4 | 103.0 |
| 64 | 103.5 | **107.8** | 102.8 | 102.5 | 104.1 | 103.4 | **107.5** | 102.7 | 103.4 | 102.1 |

Table 4: Attention layer speedups at different sparsity levels, on MNLI and QQP.

| | Speedup(QNLI) (%) | | | | | Speedup(SST-2) (%) | | | | |
|---|---|---|---|---|---|---|---|---|---|---|
| K | PASS | PASSCONC | DSP | LAG | Voita | PASS | PASSCONC | DSP | LAG | Voita |
| 16 | 113.8 | **143.7** | 118.3 | 127.5 | 138.8 | 111.5 | **163.3** | 148.8 | 107.9 | 107.2 |
| 32 | 105.1 | **125.6** | 103.1 | 108.2 | 107.9 | 102.2 | **136.7** | 106.9 | 102.2 | 102.8 |
| 48 | **103.8** | 103.5 | 103.1 | 103.5 | 102.8 | 101.9 | **107.2** | 102.5 | 103.1 | 102.5 |
| 64 | 103.8 | **107.5** | 103.1 | 103.1 | 103.1 | 102.2 | **102.8** | 102.8 | 102.8 | 102.8 |

Table 5: Attention layer speedups at different sparsity levels, on QNLI and SST-2.

| | Accuracy(QNLI) (%) | | | | | Accuracy(SST-2) (%) | | | | |
|---|---|---|---|---|---|---|---|---|---|---|
| K | PASS | PASSCONC | DSP | LAG | Voita | PASS | PASSCONC | DSP | LAG | Voita |
| 16 | **86.03** | 84.79 | 84.59 | 83.49 | 84.90 | **90.48** | 89.33 | 83.37 | 86.81 | 88.76 |
| 32 | **88.43** | 85.83 | 88.38 | 87.50 | 87.59 | 91.28 | 91.06 | **92.32** | 90.14 | 90.71 |
| 48 | 89.13 | 88.80 | **89.58** | 89.13 | 88.91 | **92.32** | 91.51 | 91.40 | 90.48 | 90.94 |
| 64 | 89.57 | 89.51 | 89.68 | **89.69** | 82.61 | 91.28 | 91.17 | **91.74** | 90.14 | 91.28 |

Table 6: Subnetwork performance at different sparsity levels, on QNLI and SST-2.

| | Accuracy(MNLI) (%) | | | | | Accuracy(QQP) (%) | | | | |
|---|---|---|---|---|---|---|---|---|---|---|
| K | PASS | PASSCONC | DSP | LAG | Voita | PASS | PASSCONC | DSP | LAG | Voita |
| 16 | 79.38 | 78.38 | **80.69** | 78.19 | 80.14 | 89.18 | 88.50 | **89.25** | 86.87 | 85.53 |
| 32 | 80.73 | 79.63 | **82.64** | 81.33 | 80.30 | 89.94 | 89.36 | **90.09** | 88.99 | 89.99 |
| 48 | 82.59 | 82.51 | **82.92** | 82.80 | 82.03 | **90.57** | 90.40 | 90.16 | 89.67 | 90.20 |
| 64 | 83.23 | 83.06 | 83.55 | **83.74** | 75.70 | **90.80** | 90.74 | 90.28 | 90.02 | 87.21 |

Table 7: Subnetwork performance at different sparsity levels, on MNLI and QQP.

