# OpenReview forum: "Pruning Attention Heads with Almost-sure Sparsity Targets"
_ICLR.cc/2024/Conference — Submitted to ICLR 2024_

### Official Review · Reviewer_WBEj · 2023-10-29

**Soundness:** 2 fair
**Presentation:** 2 fair
**Contribution:** 2 fair
**Rating:** 3
**Confidence:** 4

**Summary:**

This work introduces the concept of almost-sure sparsity for attention-head-pruning of language models. The gate parameters are jointly trained with the main network parameter, under the proposed PASS loss with the gate closing, gate opening, and non-binary probabilities. Furthermore, the proposed Concentrator loss ensures that the retained attention heads are primarily concentrated in a limited number of layers, leading to better latency improvements.

**Strengths:**

- Given the popularity of Large Language Models (LLMs) and Vision Transformers (ViTs), the compression of transformer-based models has emerged as a topic of significant interest.
- Considering existing structured pruning works that overlook the latency, I appreciate the proposed 'Concentrator', which boosts inference speedups.

**Weaknesses:**

- The original networks chosen for pruning (i.e., a 6-layer transformer and a 12-layer BERT (perhaps BERT-base?)) seem outdated. Could its applicability be demonstrated on LLMs (at least, 7B-LLaMA)? A potential approach might involve integrating or contrasting the proposed work with LLM-Pruner (https://arxiv.org/abs/2305.11627).
- The performance gains in Table 1 are not particularly impressive, considering additional hyperparameters. How did the authors choose the value of lambda and lambda_c in (9)? Why was CoFi-Pruning (https://arxiv.org/abs/2204.00408) not included in the comparison?
- Analyzing which parts are pruned would be interesting, to understand the network behavior.
- Please include the number of parameters and the performance of the original network.

**Questions:**

Please see the Weaknesses part.

---

### Official Review · Reviewer_BTZ5 · 2023-11-04

**Soundness:** 3 good
**Presentation:** 4 excellent
**Contribution:** 3 good
**Rating:** 5
**Confidence:** 4

**Summary:**

This paper proposes a novel method to prune the attention heads in a Transformer model which results in impressive speedups in the attention layer during inference. The proposed method, PASS (Pruning with Almost Sure Sparsity targets) jointly learns the distribution of gates acting on top of each of the attention heads during the training / fine-tuning stage. The gates control whether a specific attention head will be pruned or not. To improve the inference efficiency further, the paper uses a concentrator loss to reduce the number of layers with unpruned heads, developing the PASSCONC (PASS with concentrator) framework. Clipping and selective re-opening strategy of closed gates during training results in better optimization and makes the proposed approach work well with respect to the speedup. The speedup gains on translation task using encoder-decoder architecture and on BLUE tasks using BERT are impressive.

**Strengths:**

- The paper tackles an important problem of pruning models to obtain sparser architectures that are fast during inference, especially given the large scale deployment of Transformer-based models.
- The proposed approaches - PASS and PASSCONC are novel in my opinion, and one of the only approaches that don't have to use a threshold probability for the gates that are being trained because of the almost sure sparsity constraints during training.
- The paper builds the theory coherently by first building the $\mathcal{R_{pass}}$ objective that avoids the local optimum issues of the $\mathcal{R_{base}}$ loss objective. Subsequently, $\mathcal{R_{passconc}}$ is proposed that reduces the number of layers with unpruned heads resulting in further speedups over PASS. Finally, clipping of $\phi_{i}$'s to avoid zero gradients and selective reopening of closed gates using confidence scores improves the training process.
- The empirical results on IWSLT translation dataset and the four GLUE benchmark tasks are impressive compared to the baselines.

**Weaknesses:**

- A couple of claims in the introduction are wrong in my opinion. Firstly, several followup works to [1] and [2] have shown that attention is not really interpretable, for example see [3]. Secondly, the 50% number for inference latency in the attention layer is highly dependent on the model size and sequence length used for the model. For instance, a 1.5B parameter transformer model with 2k sequence length can have only around 25% of its computation being spent in the attention layer. For larger models, this percentage is reduced further. This is because the feedforward layer has huge weight matrices that are sharded across multiple accelerators which results in huge communication costs. I believe this blanket 50% number line should be removed.
- The notion of almost-sure sparsity is not exactly novel and the first contribution listed in the paper feels bloated. The words used in other works might be different but sparsity research on skewing the probability distribution towards either 0 or 1 does exist.
- I believe a proof/derivation is required for the claim of the gradient descent direction of $\mathcal{R_{base}}$ mentioned above equation 7 on page 5. Can the authors please add this proof in the appendix?
- In my opinion, the accuracy/BLEU score results should be reported for the unpruned model baseline too. This would give an idea of the difference between the pruned and the unpruned models.
- Why did the authors only report numbers on 4 tasks in the GLUE benchmark, given that it is composed of 10 tasks. WNLI and AX have very high variances, so they can be skipped, but the numbers should be reported for the other 4 tasks too.
- The empirical section is a bit weak. How does the proposed approach scale to larger models and decoder-only models? Can the authors at-least report performance of some QA tasks with large sequence length like HotpotQA or summarization tasks on datasets like Pubmed/arxiv etc.
- Some important ablation studies comparing PASS and PASSCONC are missing like how many layers have unpruned attention heads in each of the experiments with the introduction of the concentrator loss?

[1]: Voita et al., Analyzing multi-head self attention: Specialized heads do the heavy lifting, the rest can be pruned, 2019.

[2]: Clark et al., What does BERT look at? an analysis of BERT's attention, 2019.

[3]: Pruthi et al., Learning to Deceive with Attention-Based Explanations, 2020.

**Questions:**

I have asked most of my questions in the weakness section, but here is one additional suggestion: Please add the meaning of each datapoint in Figure 6. I can infer that it represents the value $K$ (top-$K$ attention heads), but it requires context to figure out. Also add the datapoint on the accuracy of the unpruned model in Figure 6.

If the authors can address the questions in the weakness section and incorporate some of the suggestions, I am willing to increase my score.

---

### Official Review · Reviewer_fbDA · 2023-11-04

**Soundness:** 3 good
**Presentation:** 3 good
**Contribution:** 3 good
**Rating:** 6
**Confidence:** 4

**Summary:**

This paper proposes a generic framework for pruning with almost-sure sparsity with Hard Concrete distribution, which also mitigate the train-testing inconsistency.
For further speedup, this paper also introduces a concentrator regularization to facilitate entire head pruning at some layers.
The proposed approach achieves competitive model acceleration while maintain comparable latency / BLEU scores, over the leading approaches, based on experiments on encoder-decoder (ED) Transformer and BERT.

**Strengths:**

1. The proposed PASS and PASSCONC framework is effective in pruning away the redundant heads.
2. This paper also introduce the theoretical analysis on the formulation of PASS and PASSCONC.
3. Especially at the high sparsity level, PASS and PASSCONC provide significant speedup while preserving competitive accuracy / BLEU scores.

**Weaknesses:**

1. Clipping and selective reopening strategy are an important part to facilitate efficiency improvement. But it's not clear how these are implemented. Could you add some details or formulate the strategy?
2. It's not clear what's the impact of the training-testing inconsistency.

**Questions:**

1. How would the proposed approach scale to large and deeper layers?
2. What are the compute overheads with proposed approach as compared to the simpler yet effective DSP framework?

---

### Official Review · Reviewer_K7dC · 2023-11-08

**Soundness:** 3 good
**Presentation:** 3 good
**Contribution:** 2 fair
**Rating:** 5
**Confidence:** 3

**Summary:**

The paper proposes a regularizer that induces prunable attention heads in transformer models. The regularizer encourages the attention heads' gating variables to have high probabilities of taking binary values, 0/1. This regularizer is applied jointly with the training loss of transformers. Moreover, the authors extend the framework by encouraging all heads in a layer to be prunable, leading to skippable attention blocks, and further reduced latency.

Experiments compare against several other regularizers, and seem to suggest the proposed regularizer is the best.

**Strengths:**

1. The paper makes some connection between learning the gating variables and evidence lower bound, which seems interesting.
2. The illustrative analysis on why adopting Eq (7) instead Eq (6) is interesting.

**Weaknesses:**

1. In terms of methodology, (Voita et al 2019) also uses hard concrete distribution to model the probability of a closed gate. The only difference is in the form of regularizer. It is less clear why the proposed regularizer is better than Voita's, though empirically it seems so. With that been said, novelty of this paper is also limited.

2. Other than just reporting test metrics, the author should show the pattern of learned gate variables, and draw some insights there. This is especially important for showing the difference against (Voita et al 2019). Also, how concentrated are the prunable heads due to PASSCONC? Are they distributed at the bottom, middle, or top of the transformer layers?

3. The clip and re-open trick seems a very crucial trick. Could it also be applied to (Voita et al 2019), would that also bring some gain on (Voita et al 2019)'s method?

**Questions:**

See weakness 2 and 3.

Another question, (Voita et al 2019)'s method doesn't specify sparsity level directly. So how to make sure the method yield a pruning pattern with pre-defined sparsity level? Footnote 5 isn't self-contained enough.

---

### Meta-Review · Area_Chair_z9p4 · 2023-12-10

**Metareview:**

The paper proposes a method for pruning attention heads in Transformer models by enforcing the so called almost-sure sparsity on the mask/gate variables during training. To further improve efficiency, it adds a concentrator loss to reduce layers with unpruned heads.

Strengths:
- Good theoretical analysis connecting the formulation to evidence lower bounds and learning with latent variable models.
- Good speedups, especially at high sparsity levels, while maintaining accuracy.

Weaknesses:
- The difference from prior work like Voita et al. is not very clearly explained. The novelty seems somewhat limited.
- Some implementation details like clipping or reopening are not provided.
- The implications are note widely applicable to larger and deeper models.

Rooms for improvements:
- Better differentiate with prior work from Voita et al.
- Provide more implementation and ablation details, e.g. more exploratory analysis on the learned gates or the approximate surrogate distributoin.
- Demonstrate scaling up to larger models and on borader NLP tasks

**Justification For Why Not Higher Score:**

Lack of novelty and clear distinction with prior work and lack of generality and intuitive explanations of the findings.

**Justification For Why Not Lower Score:**

N/A

---

### Decision · Program_Chairs · 2024-01-16

Reject